# The Application and Challenges of Brain Organoids in Exploring the Mechanism of Arbovirus Infection

**DOI:** 10.3390/microorganisms13061281

**Published:** 2025-05-30

**Authors:** Baoqiu Cui, Zhijie Wang, Anum Farid, Zeyu Wang, Kaiyue Wei, Naixia Ren, Fengtang Yang, Hong Liu

**Affiliations:** Center for Organoid and Genome Research, School of Life Sciences and Medicine, Shandong University of Technology, Zibo 255049, China; cuibaoqiu1314@hotmail.com (B.C.); wangzhijie0607@hotmail.com (Z.W.); anum.malik911@gmail.com (A.F.); wangzeyu0701@hotmail.com (Z.W.); wky24410011008@outlook.com (K.W.); fengtangyang@163.com (F.Y.)

**Keywords:** brain organoids, arboviruses, Japanese encephalitis virus, Zika virus, La Crosse virus, mechanisms of infection, immune response

## Abstract

Arboviruses, transmitted by blood-sucking arthropods, are responsible for significant human and animal diseases, including fever, hemorrhagic fever, and encephalitis, posing a serious threat to global public health. Nevertheless, research on the mechanisms of arbovirus infection and the development of therapeutic interventions has been impeded. This delay is primarily due to the limitations inherent in current in vitro research models, including cell cultures and animal models. The simplicity of cell types and interspecies differences present significant obstacles to advancing our understanding of arbovirus infection mechanisms and the development of effective drugs. Human brain organoids, derived from human pluripotent stem cells or human embryonic stem cells and cultured in three-dimensional systems, more accurately replicate the extensive neuronal cellular diversity and key characteristics of human neurodevelopment. These organoids serve as an ideal model for investigating the intricate interactions between viruses and human hosts, and providing a novel platform for the development of antiviral drugs. In this review, we summarize how brain organoid models complement classical approaches to accelerate research into the infection mechanisms of arboviruses, with a particular focus on the types of neural cells, key factors, and cellular signaling pathways involved in the arbovirus infection of brain organoids that have been reported. Furthermore, we examine the development of brain organoids, address their current limitations, and propose future directions to enhance the application of brain organoids in the study of arboviral infectious diseases.

## 1. Introduction

Arboviruses are a diverse group of viruses transmitted by blood-sucking arthropods, such as mosquitoes and ticks, causing severe diseases in both humans and animals [1]. More than 500 arboviruses have been identified, of which more than 100 can cause human and animal diseases [2]. Arboviral infections account for approximately 17% of all infectious diseases worldwide, causing 1 billion cases and 1 million deaths annually [3,4]. Most arbovirus infections do not cause neuroinvasive disease, but some arbovirus families pose a major global health threat due to their ability to cause rapid outbreaks and severe conditions like encephalopathy, meningoencephalitis, myelitis, and Guillain–Barré symptoms (GBS) [5]. These neuroinvasive arboviruses predominantly belong to the families *Flaviviridae*, *Bunyaviridae*, *Togaviridae*, and *Reoviridae* [6]. The *Flaviviridae* family includes Japanese encephalitis virus (JEV), Zika virus (ZIKV), dengue virus (DENV), and West Nile virus (WNV) [6]. The *Bunyaviridae* family comprises California encephalitis virus (CEV), Rift Valley fever virus (RVFV), and La Crosse virus (LACV) [7]. The *Togaviridae* family consists of Eastern equine encephalitis virus (EEEV), Western equine encephalitis virus (WEEV), Venezuelan equine encephalitis virus (VEEV), and Chikungunya virus (CHIKV) [8]. Lastly, the *Reoviridae* family includes the Colorado tick fever virus (CTFV) [6].

Arboviruses persist in natural ecosystems through a diverse array of vectors and hosts, rendering their complete eradication exceedingly challenging. Demonstrating the ongoing threat of these viruses, multiple outbreaks and epidemics have been reported in the past two decades (Figure 1) [9,10,11,12,13,14,15,16,17,18]. Except for JEV and yellow fever virus (YFV), effective vaccines for humans are lacking for other arboviruses, and there are currently no effective treatments available for arboviral diseases. To address this issue, it is essential to strengthen the surveillance and research on arboviruses, gain a deeper understanding of their pathogenic mechanisms, and advance the development of vaccines and therapeutics to mitigate the risks posed by arboviruses.

Traditional in vitro models for studying arbovirus infection mainly include two-dimensional (2D) cultured cell line models (such as BHK cells, Vero cells, C6/36 cells) and animal models (such as mice, rats, monkeys and transgenic animal model. However, these in vitro models have three major limitations that are difficult to overcome: (1) species differences, (2) non-neural cell origin (2D culture systems), (3) single cellular type (2D culture systems), which hinder the in-depth understanding of arbovirus infection mechanisms, and the development of drugs and vaccines. First, whether it is a 2D cultured cell line or an animal model, it is difficult to accurately replicate human-specific conditions due to genetic differences that lead to variations in physiological functions, immune responses, and pathophysiological states. These species-specific differences often lead to discrepancies when translating research findings into clinical applications. For example, interferon gene-stimulating protein (STING) plays a crucial role in innate immunity, but human STING (hSTING) proteins differ significantly in structural conformation from mouse STING (mSTING) proteins. These species’ specific differences result in ligand drugs interacting with STING proteins exhibiting different binding efficiencies and functional activities in mouse models and humans. Genetic heterogeneity results in inconsistencies between animal models and humans in disease transmission pathways, disease progression, and clinical symptoms [19,20]. In addition, human immune factors expressed in transgenic mice can produce immune cross-reactions with mice, resulting in unexpected differences in disease phenotypes [21]. Secondly, 2D cell lines (such as BHK cells and Vero cells) commonly used in arbovirus research are mostly non-neural cells, while the main target of arbovirus is the nervous system. These traditional cell lines cannot simulate the infection process of the virus on nerve cells, nor can they reflect the replication, transmission and pathogenesis mechanism of virus in nerve tissues, thus limiting the in-depth study of the neural invasion and neurotoxicity mechanism of virus. Third, cell lines in a 2D culture system are usually single cellular types, lacking multi-cell synergy and complex tissue structure [22]. The infection process of arbovirus involves multiple interactions between virus and host cells, immune system and microenvironment, while the 2D culture system cannot simulate this three-dimensional system (3D) tissue structure and dynamic cooperation between cells, so it is difficult to truly reflect the complexity of virus–host interaction in vivo.

## 2. Development of Brain Organoids

Brain organoids are 3D cultures containing multiple neural cell types formed by the induced differentiation of human embryonic stem cells (hESC) or human-induced pluripotent stem cells (hiPSCs) into self-organizing tissues. These self-organizing tissues mimic aspects of human brain development and function under in vitro conditions [23,24]. In 2001, hESCs aggregated and differentiated into embryoid bodies in vitro, and formed neural tube-like structures under the action of fibroblast growth factor 2 (FGF-2) [25]. In 2005, mouse embryonic stem cells were cultured in serum-free suspension, and the embryoid bodies produced by them were differentiated into telencephalon [26]. In 2016, Qian et al. engineered a miniature bioreactor, termed SpinΩ, for the generation of human brain organoids [23]. Through the stage-specific modulation of signaling pathways, they successfully simulated human brain development in vitro. During the initial 1–7 days of culture, the bone morphogenetic protein (BMP) signaling pathway inhibitor, Dorsomorphin, and the Transforming Growth Factor-beta (TGF-β) inhibitor, A83-01, were administered to facilitate the efficient directed differentiation of neural precursor cells via dual-pathway inhibition. Between days 7 and 14, the Wnt pathway activator CHIR99021 (or WNT3A protein) was administered in conjunction with the TGF-β inhibitor SB431542 to synergistically promote the induction of neuroectoderm. Between days 14 and 71, N2/B27 supplements were employed to fulfill the fundamental requirements for neural development. Post day 71, the culture medium was enhanced to include Neurobasal and B27, along with Brain-Derived Neurotrophic Factor (BDNF), Glial Cell Line-Derived Neurotrophic Factor (GDNF), Transforming Growth Factor Beta (TGF-β), and cyclic Adenosine Monophosphate (cAMP) to facilitate the functional maturation of neural networks. Commencing on day 14 of culture, under the dynamic conditions provided by the SpinΩ bioreactor, the organoids underwent a sequential differentiation into distinct yet interdependent brain region structures. These included ventricular zone (VZ)-like and subventricular zone (SVZ)-like structures, with a cortical plate (CP)-like structure developing above the VZ and SVZ. The development of these regions parallels the process of human embryonic cortical development, as described in the reference [23]. Dang et al. utilized the hanging drop technique during the formation of embryoid bodies, incorporating basic fibroblast growth factor (bFGF), non-essential amino acids (NEAAs), and glutamine. Following a two-day period, the cultures were transferred to sterile dishes for continued cultivation [24]. During the neural induction phase, the addition of N2, NEAA, glutamine, and heparin facilitated the differentiation of neural precursor cells. By day 11, the brain organoids were transferred to Matrigel and supplemented with B27 supplements without vitamin A, N2, NEAA, insulin, beta-mercaptoethanol, and glutamine to facilitate maturation and regionalized development. On day 15, the organoids were cultured in stir flask bioreactors with the addition of retinoic acid and vitamin A, with medium changes every three days for a long-term dynamic culture. Ultimately, early developmental features of the forebrain, hippocampus, dorsal cortex, and internal neuronal regions were observed in the brain organoids. Calcium imaging confirmed the regionalization, cortical differentiation, and functional neural electrical activity of the organoids [24]. Through the transcriptomic analysis of brain organoids and human brain tissue, the study revealed that in vitro differentiated brain organoids share a high degree of similarity with early fetal brain tissue. Notably, 100-day-old brain organoids demonstrated the strongest correlation with fetal brain tissue, effectively modeling the brain development of early pregnancy [23,24]. In 2017, Watanabe et al. added ventral analog signals during the culture of brain organoids to produce basal ganglionic eminence (GE)-like structures, resulting in GE organoids [27]. Figure 2 illustrates the key time nodes of brain organoid development and maturation in our laboratory. Embryoid bodies (EBs) were gradually formed in 0~5 days, which were smooth and round, with a diameter of >300 μm; at about 7 days, EBs showed a round edge translucent shape, and then entered the amplification stage of neuroepithelial cells. The EBs surface sprouted. After 10 days, brain organoids gradually matured. At 40 days, the center of brain organoids was dense and the edge was translucent. In recent years, advancements in brain organoid technology have facilitated the differentiation of various types of brain organoids, including forebrain organoids [23,28], cortical organoids [29], and choroid plexus organoids [30]. Furthermore, brain organoids can be co-cultured with other cell types, such as microglia [31], the resident immune cells of the central nervous system (CNS), to simulate a more complex system environment and replicate the immune microenvironment.

## 3. Application of Brain Organoids in the Study of Arbovirus Infection Mechanism

Brain organoids have now been used to study a variety of viral infections that infect the nervous system, such as JEV [29], ZIKV [23,24,28,32,33,34,35,36,37,38,39,40,41,42], and La Crosse virus (LACV) [43] (Table 1 and Figure 3).

In studies of arbovirus infections (e.g., ZIKV, JEV, LACV), brain organoids derived from hESCs and hiPSCs are used equally often (hESC: 8 times; hiPSC: 9 times), with no preference for either cell type. Brain organoids featuring multiple brain regions are the most common model, using a culture system of DMEM/F12 with Neurobasal medium, N2/B27 supplements, and SMAD pathway inhibitors (SB-431542 or LDN-193189) with bFGF/EGF for neural induction. During maturation, neurotrophic factors like BDNF and GDNF, combined with a Matrigel 3D culture system, support synapse formation and neuronal maturation. Rotary [23,32,37,42] and swinging [34,36,43] oscillation cultures are preferred for better nutrient diffusion and oxygenation (Appendix A).

### 3.1. ZIKV Infection Model

ZIKV is an enveloped RNA flavivirus mainly transmitted by Aedes aegypti, which can cause neonatal microcephaly [32,38,44] and GBS [45].

ZIKV has diverged into two distinct lineages: the Asian lineage and the African lineage. The African lineage is generally not associated with the onset of microcephaly [46], whereas the Asian lineage has a strong correlation with microcephaly syndrome. In 2D neural cultures, both the African prototype strain MR766 and Asian lineage strains (H/PF/2013 and FB-–GWUH-–2016) are capable of infecting NPCs. In 3D brain organoids, the African strain MR766 demonstrates the most significant inhibitory effect on NPC proliferation, primarily localizing to the primitive cortical region of brain organoids and targeting surface-layer immature neurons, intermediate progenitor cells, and astrocytes. Conversely, the Asian lineage strains (H/PF/2013 and FB-GWUH-2016) readily target to and replicate in proliferating VZ apical progenitors. The main phenotypic effect was the premature differentiation of neural progenitors associated with centrosome perturbation, even during early stages of infection, leading to progenitor depletion, the disruption of the VZ, impaired neurogenesis, and cortical thinning [40,47,48]. Moreover, ZIKV can infect NPCs, cortical progenitor cells, oRGCs, etc. These types of nerve cells play an indispensable role in brain development. Because ZIKV infection leads to the death of these nerve cells, the number of nerve cells is reduced, which eventually leads to the atrophy of brain organoids [23,24,38,39,49,50].

ZIKV infection activates TLR3, and using TLR3 inhibitors, can reduce apoptosis and brain organ atrophy [24]. ZIKV produces sfRNA that affects brain organoid development through the Wnt signaling pathway and pro-apoptotic pathways [41]. RNAi can inhibit ZIKV replication and reduce damage [39].

In antiviral studies, various compounds have shown different mechanisms of action against ZIKV, such as 25HC blocking ZIKV internalization [33], enoxacin effectively degrading viral RNA [39], and BA reducing NPCs apoptosis [42]. However, co-treatment with STX and ZIKV aggravates NPC death and nervous system damage [36].

### 3.2. JEV Infection Model

JEV is a significant cause of viral encephalitis in Asia, with a high incidence rate and severe outcomes, including neurological sequelae and death. The virus is primarily transmitted through mosquito bites, particularly by the Culex species. Pigs and birds serve as amplifying hosts, contributing to the spread of the virus in endemic regions. The disease poses a substantial public health challenge, especially in areas with high mosquito activity and inadequate vaccination coverage [51,52].

Zhang et al. [29] used JEV to infect the cortical organoids derived from hESC. The development of the brain organoids is similar to that of the human cortex. During the culture of the brain organoids, multiple brain regions and multiple neuronal subtypes gradually emerged. After JEV (SA14 strain) infection, with the increase in the virus titer, the number of proliferating cells decreased and the neuronal layer became thinner. JEV tends to infect NPCs and oRGCs in cortical organoids, but it is more inclined to infect astrocytes in brain organoids over 100 days. However, the early brain organoids did not cause the secretion of IFN-β, and the early brain organoids may be more suitable for exploring the mechanism of virus infection and replication in the brain organoids. It is worth noting that, unlike ZIKV, JEV infection does not activate the innate immune receptor TLR3, suggesting that JEV may have a unique mechanism to evade or inhibit innate immune responses. In addition, JEV infection can effectively upregulate the expression of RIG-I and induce the expression of IFN-β, and then activate the expression of interferon-stimulated genes (ISGs), which provides new clues for understanding the pathogenesis of JEV [29].

### 3.3. LACV Infection Model

LACV, an arbovirus primarily endemic to the Midwestern and mid-–Atlantic regions of the United States, is capable of inducing encephalitis and presents a particularly significant risk to pediatric populations [53,54]. Clayton employed human cerebral organoids (COs) derived from iPSCs to investigate LACV infection. These organoids encompass neurons at different developmental stages and exhibit internal structures and regional organization analogous to those of the developing human brain. They are capable of replicating the neurogenic program characteristic of normal human cortical development and consist of diverse neural lineage cell types that deliver crucial trophic and developmental signals. In comparison to monolayer cell cultures, COs provide a more precise representation of the infection dynamics of LACV in the developing human brain.

Contrary to the previously held belief that differentiated neurons exhibit greater resistance to LACV-induced apoptosis, this study demonstrates that developmentally committed neurons, are more susceptible to LACV-induced apoptosis than NPCs [43]. This finding suggests that neuronal responses to viral infection in complex neural tissues differ from those observed in monolayer cell cultures. Furthermore, the study reveals that NPCs exhibit more robust IFN signaling and ISGs responses to LACV, whereas committed neurons display weaker responses—an observation that contrasts with previous studies on differentiated neurons in monolayer cultures. The exogenous administration of recombinant IFN was found to enhance cell viability in LACV-infected COs, particularly by promoting the survival of committed neurons. These results suggest a novel therapeutic strategy for viral encephalitis, indicating that the enhancement of IFN signaling may be crucial in preventing virus-induced neuronal death.

While ZIKV, JEV, and LACV all exhibit neurotropism, their neural infection mechanisms differ significantly. ZIKV primarily impairs neurogenesis through TLR3-mediated apoptosis and sfRNA interference with the Wnt pathway, demonstrating strain-specific tropism. The Asian strain predominantly targets apical progenitor cells, whereas the African strain shows a preference for cortical neurons. In contrast, JEV, despite not activating the TLR3 pathway, markedly upregulates RIG-I and IFN-β, thereby initiating the ISG response and activating the host’s innate immune response. Both ZIKV and JEV predominantly target NPCs and oRGCs, whereas LACV exhibits a stronger affinity for committed neurons, whose interferon response is weaker compared to that of NPCs. This variation in susceptibility underscores the critical role of developmental stages in viral infection.

## 4. The Limitation of Brain Organoids in Arbovirus Research: Challenges and Implications

Brain organoids, 3D models derived from hESCs or hiPSCs, have emerged as powerful tools for studying arbovirus infections and their effects on the developing brain. However, several limitations persist in utilizing brain organoids for arbovirus research, (1) Brain organoid model heterogeneity, (2) immunodeficient state of brain organoid, (3) short lifespan of brain organoid viability.

### 4.1. Heterogeneity of Brain Organoids

“Batch-to-batch” heterogeneity remains a major challenge in brain organoid-based arbovirus studies (ISCO White Paper, 2023). The differentiation of brain organoids from hESCs or hiPSCs is influenced by multiple factors, including the origin of stem cells, differentiation protocols, and culture conditions. These parameters collectively contribute to substantial interlaboratory variability in both the developmental timeline and structural organization of the resulting brain organoids. Notably, while certain protocols yield neural tube-like structures within 14 days [29], others require extended differentiation periods up to 35 days [23]. This temporal disparity primarily reflects protocol-dependent variations in the differentiation kinetics of NPCs and subsequent neuroepithelial morphogenesis. The susceptibility of brain organoids to arboviral infection exhibits strong dependence on their structural maturity and developmental stage. Organoids exhibiting well-defined ventricular VZ and SVZ recapitulate the cellular microenvironments preferentially targeted by neurotropic arboviruses, thereby providing more biologically faithful models for investigating viral tropism. Conversely, immature organoids frequently lack the necessary cytoarchitectural complexity to adequately model the complete viral life cycle, including initial infection, intracellular replication, and intercellular spread. This developmental and architectural heterogeneity underscores the critical need to establish standardized culture protocols to enhance experimental reproducibility and ensure model consistency.

### 4.2. Immunodeficiency in Brain Organoids

Brain organoids inherently exist in an immunodeficient state due to the absence of resident immune cells (particularly microglia) and circulating immune components. The immunodeficient status impacts the developmental progression and structural integrity of brain organoids and significantly limits their ability to model host immune responses to viral infections and neuro inflammation. The immune system plays a pivotal role in combating viral infections, particularly in the CNS [31,55,56,57]. Immunodeficiency impairs the viral clearance capacity of brain organoids, thereby affecting the accuracy and reliability of research outcomes. Furthermore, the viral infection can disrupt neuronal and microglial functions, leading to structural abnormalities and developmental impairments in brain organoids [55,56,57]. Immunodeficiency may exacerbate these effects, preventing the brain organoid model from accurately reflecting the true pathophysiological state of the organism post-viral infection. Additionally, immunodeficiency may alter viral infection routes and mechanisms, consequently affecting our understanding of viral pathogenic characteristics [58].

### 4.3. Short Lifespan of Brain Organoids

One of the primary reasons for the short lifespan of brain organoids is the lack of a vascular system, which limits nutrient and oxygen supply to the inner regions of the organoid, leading to necrosis and reduced viability over time. Another contributing factor is the inherent variability in organoid cultures, which can lead to inconsistencies in their development and maturation. This variability can result from the differences in the protocols used for organoid generation, as well as from the genetic background of the stem cells used. Studies have shown that organoids can exhibit high variability in recapitulating primary tissue, which can affect their lifespan and functionality [59].

Additionally, the lack of certain cell types, such as microglia [31,55,56,57], in many brain organoid models, can impact their development and longevity [60]. Microglia play crucial roles in brain development and homeostasis, and their absence can lead to the incomplete modeling of brain physiology [61]. Moreover, the limited maturation of neurons within organoids can also contribute to their short lifespan. Neurons in organoids often fail to reach the same level of maturity as those in vivo, which can affect their long-term viability and functionality [62]. Finally, the lack of a supportive microenvironment, including signaling gradients and mechanical cues, can hinder the development and longevity of brain organoids.

### 4.4. Epistemological Limitations

At present, brain organoids remain at a developmental stage analogous to that of a fetus and are not yet capable of accurately replicating the physiological and pathological characteristics of the adult brain. This limitation includes the inability to model the mechanisms of arbovirus infections, such as those caused by the JEV, within a mature nervous system. Furthermore, the absence of vascularization and an immune microenvironment in these brain-like structures impedes the investigation of cross-organ interactions, such as the brain–gut axis, and hinders the evaluation of systemic inflammation and immune responses on the nervous system.

### 4.5. Ethical Controversy

Brain organoid models, despite offering significant advantages in elucidating the mechanisms of arbovirus neuroinvasion, continue to encounter numerous challenges and limitations. From a technical standpoint, current models possess inherent deficiencies, including the heterogeneity of cell types, the absence of vascularization and immune microenvironments, and limited culture duration. In terms of ethical considerations, this nascent field is confronted with several unresolved dilemmas. The ethical debate surrounding the use of hESCs and the complexities of obtaining informed consent for hiPSC samples, particularly in cases involving surrogate decision making for cognitively impaired patients, establish significant ethical barriers to research [63,64], The transplantation of virus-infected cerebral organoids into animal models for the study of the blood–brain barrier presents significant ethical challenges regarding the ‘humanization’ of animals and heightens the biosafety risk associated with the potential cross-species transmission of pathogens [63]. With the progression of cultivation technology, the prolonged cultivation of brain organoids may raise contentious issues regarding the potential for consciousness and their moral status. Consequently, there is an urgent need to develop a rigorous scientific framework for evaluating the ‘threshold of consciousness’ [64,65,66,67]. Arbovirus research demands strict biosafety due to its high pathogenicity, while the genetic data in personalized organoids raise donor privacy concerns. These ethical issues, along with technical and translational medicine challenges, form the complex barriers in brain organoid research on arboviruses.

## 5. Future Directions

Brain organoids, as an innovative in vitro model for researching arboviral infections, confront several intricate challenges, especially concerning model heterogeneity, immunodeficiency, and their limited duration of viability. To mitigate these challenges, future research should prioritize the following critical areas.

### 5.1. Establish a Standardized Culture System

Establishing a standardized culture system is of paramount importance to address the impacts of variations in species origins, experimental batches, and inter-laboratory discrepancies on the reproducibility of cerebral organoid research outcomes. To address this challenge, it is essential to optimize differentiation protocols by systematically comparing culture and differentiation processes across laboratories, identifying critical factors influencing NPCs’ differentiation and neuroepithelial formation, and subsequently establishing reproducible standardized protocols based on these determinants. These benchmarks will ensure biological consistency across organoid batches, enhance experimental reproducibility and reliability, and provide a robust platform for arboviral infection research.

### 5.2. Engineering the Immune Microenvironment and Modeling the BBB

To address the immunological deficiencies and limitations in the structural–functional integrity of brain organoids, future strategies could involve the establishment of multicellular co-culture systems or the application of the CRISPR-based gene editing to induce microglial differentiation within brain organoids. This approach aims to model more physiologically relevant immune response microenvironments. Additionally, the integration of vascular endothelial cells, pericytes, and astrocytes through advanced technologies such as gene editing and 3D bioprinting could facilitate the construction of functional BBB models with selective permeability [68,69,70]. These engineered platforms would not only provide physiologically accurate models for investigating the neuroinvasive mechanisms of arboviruses but also enable the systematic exploration of virus–host immune interactions and the cascades of neuroimmune pathology following infection.

### 5.3. Constructing Vascularized Brain Organoids

The limited lifespan and maturation of brain organoids are predominantly restricted by the lack of vascular networks, resulting in inadequate oxygen and nutrient delivery to their central regions. Microfluidic technology presents an efficacious solution by allowing the precise regulation of fluid dynamics within the culture medium, thereby optimizing the delivery of oxygen and nutrients, minimizing central necrosis, and extending the viability of the organoids [69]. Furthermore, 3D printing technology can be employed to fabricate biomimetic vascular networks, offering structural support to brain organoids, facilitating vascularization, and ultimately improving their survival rate and functional maturation.

## 6. Conclusions

Arboviruses, transmitted by arthropods, represent a substantial threat to global public health. Although brain organoid technology has provided crucial insights into the mechanisms by which arboviruses such as ZIKV, JEV, and LACV disrupt neural developmental pathways and interact with host immune responses, more than 95% of pathogenic arboviruses remain systematically uncharacterized. This underscores a significant gap and untapped potential in this area of research. Progress in organoid systems, when combined with cutting-edge technologies like single-cell sequencing and spatial transcriptomics, is anticipated to facilitate a more comprehensive analysis of virus–host molecular interactions, the identification of conserved neuropathogenic patterns, and the expedited development of antiviral drugs and vaccines [71,72]. The U.S. Food and Drug Administration (FDA) made a landmark announcement in April 2025, formally endorsing organoid platforms and organoid microarray systems as novel drug screening tools. This regulatory milestone establishes their potential to replace animal models for systematic drug safety evaluation [72]. These endeavors hold the promise of delivering transformative solutions to alleviate the global health challenges posed by arboviruses.

## Figures and Tables

**Figure 1 microorganisms-13-01281-f001:**
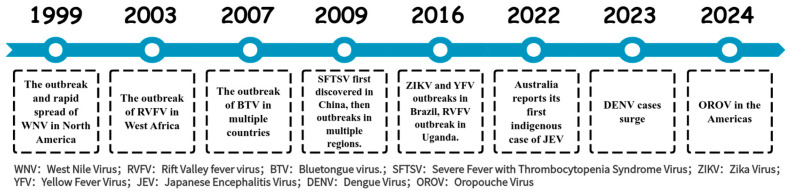
The outbreak time of arbovirus related diseases in the past two decades.

**Figure 2 microorganisms-13-01281-f002:**
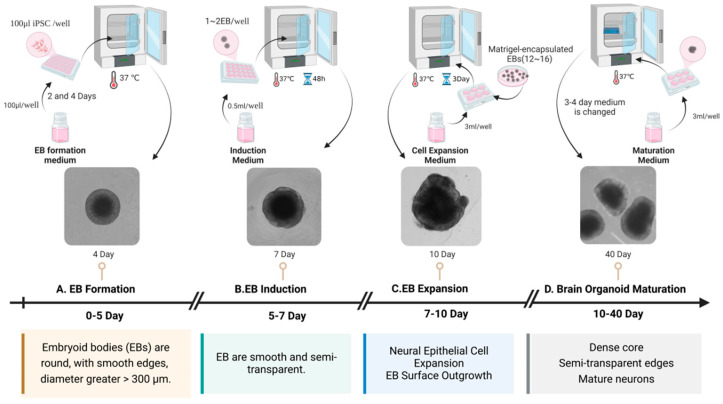
The critical temporal milestones in the development and maturation of brain organoids. This figure illustrates the process of generating brain organoids from hiPSC. (**A**) EBs are formed using an EB formation medium and cultured at 37 °C for 2–4 days, resulting in round EBs with smooth edges and diameters greater than 300 μm. (**B**) The EBs undergo induction in an induction medium, cultured at 37 °C for 5–7 days, during which the EBs become smooth and semi-transparent. (**C**) The third step involves EB expansion, where the EBs are cultured with a cell expansion medium at 37 °C for 7–10 days, promoting neural epithelial cell expansion and EB surface outgrowth. (**D**) The EBs mature into brain organoids when cultured in a maturation medium at 37 °C for 10–40 days, developing dense cores, semi-transparent edges, and mature neurons.

**Figure 3 microorganisms-13-01281-f003:**
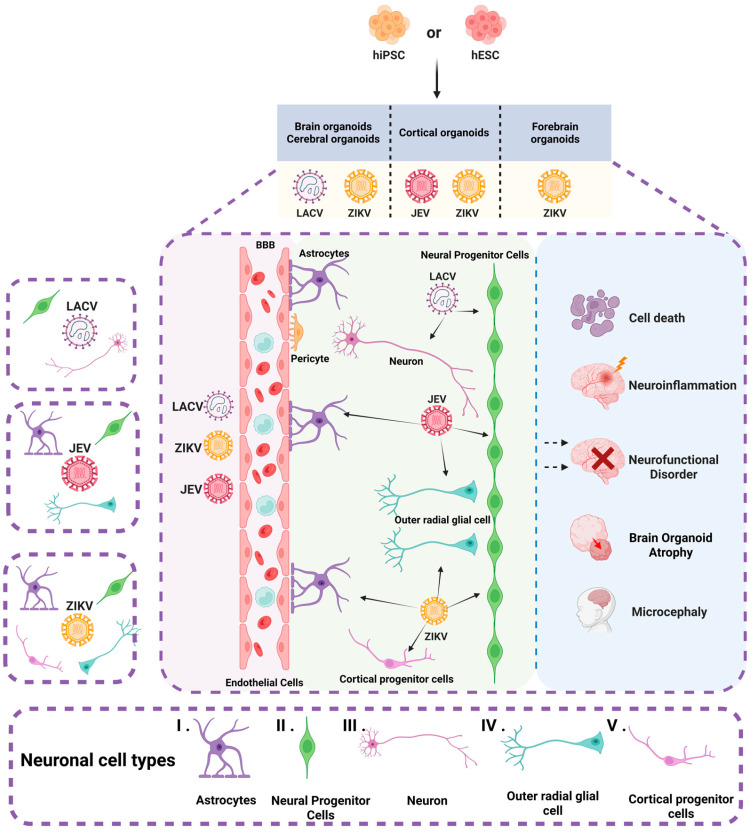
Brain organoids and neuronal cell types infected by arboviruses. The figure illustrates various types of brain organoids derived from hiPSC and hESC, including cerebral organoids, cortical organoids, and forebrain organoids. These organoids encompass a diverse array of neuronal cells, such as (**I**) astrocytes, (**II**) neural progenitor cells, (**III**) neurons, (**IV**) outer radial glial cells, and (**V**) cortical progenitor cells. JEV infects NPCs, oRGCs, and astrocytes. ZIKV predominantly targets NPCs, oRGCs, astrocytes, and cortical progenitor cells. LACV infects neurons and NPCs. Infection by these arboviruses can result in cell death, neural functional disorders, neuroinflammation, and atrophy of brain organoids. Moreover, ZIKV is specifically associated with the development of microcephaly.

**Table 1 microorganisms-13-01281-t001:** The applications of brain organoids in the investigation of arbovirus infectious mechanisms.

Virus	Disease	Origin	Types of Brain Organoids	Infected Cells	Key Factors	Key Findings	**Limitation of Brain Organoids**	**Re** **f.**
ZIKV	Microcephaly	hESC	Brain organoids *	Neural progenitor cells (NPCs) Outer radial glial cells (oRGCs)	Toll-like receptor 3 (TLR3)	Inhibition of TLR3 activity attenuates neuronal apoptosis brain-like organ contraction	II, III	[24]
ZIKV	Microcephaly	hESC	Brain organoids *	Cortical progenitor cells	/	ZIKV damages cortical areas of brain-like organs by infecting cortical progenitor cells, impairing neurodevelopment and leading to microcephaly	I, II, III	[32]
ZIKV	Microcephaly	hESC	Cortical organoids	/	25-hydroxycholesterol (25HC)	25HC inhibits ZIKV infection and protects human cortical organoid mouse embryonic brain tissue	II, III	[33]
ZIKV	Microcephaly	hESC	Cerebral organoids *	/	Monocytes	ZIKV induces monocyte migration and facilitates virus dissemination in neuronal cells	I, III	[34]
ZIKV	Microcephaly	hiPSC	Forebrain-specific organoids	NPCs oRGCs	/	ZIKV infection causes NPC death and reduced neuronal layer thickness	I, II, III	[23]
ZIKV	Microcephaly	hiPSC	Fetal-like forebrain organoids	NPCs	Hippeastrine hydrobromide (HH) Amodiaquinedihydr -ochloride dihydrate (AQ)	HH and AQ inhibit ZIKV infection and HH repairs ZIKV-induced Fetal-like forebrain organoid growth defects and differentiation	I, II	[28]
ZIKV	Microcephaly	hiPSC	Brain organoids *	/	Sofosbuvir	Sofosbuvir inhibition of ZIKV RNA polymerase attenuates replication in brain-like organs	I, II, III	[35]
ZIKV	Microcephaly	hiPSC	Brain organoids *	NPCs	Saxitoxin (STX)	STX promotes ZIKV infection of brain organoids and increases NPC death	I, II, III	[36]
ZIKV	Microcephaly	hESC	Brain organoids *	/	/	ZIKV infection alters DNA methylation in neural precursor cells, astrocytes, and differentiated neurons in human brain organoids	I, II, III	[37]
ZIKV	Microcephaly	hESC/hiPSC	Cerebral organoids *	NPCs		Severe cell death after ZIKV infection	II, III	[38]
ZIKV	Microcephaly	hESC	Brain organoids *	NPCs	RNA interference (RNAi) Enoxacin	RNAi mechanism inhibits ZIKV replication in hNPCs Enoxacin inhibits the ZIKV invasion of brain organoids	II, III	[39]
ZIKV	Microcephaly	hiPSC	Brain organoids *	NPCs	/	ZIKV strain FB-gweh-2016 and H/PF/2013 are able to cause the premature differentiation of neural precursor cells, the disruption of neurogenesis, and thinning of the cortex	II, III	[40]
ZIKV	Microcephaly	hiPSC	Brain organoids *	/	sfRNA	ZIKV sfRNA affects brain development through the Wnt signaling pathway and pro-apoptotic pathway	II, III	[41]
ZIKV	Microcephaly	hiPSC	Cerebral organoids *	/	Betulinic acid (BA)	BA inhibits neuronal cell death in ZIKV-infected brain organoids and protects the structural integrity of brain organoids.	I, II, III	[42]
JEV	Japanese encephalitis (JE)	hESC	Cortical organoids	NPCs oRGCs	Interferon (IFN) RIG-I	JEV infection can upregulate the expression of RIG-I and induce the expression of IFN-β.	I, II, III	[29]
LACV	La Crosse encephalitis (LCE)	hiPSC	Cerebral organoids *	Committed neurons	IFN	Weak interferon response in committed neurons is more sensitive to LACV	II, III	[43]

Note: /: Not available; Limitation of brain organoids: I. Lack of blood vessels; II. Lack of an immune microenvironment; III. Lack of long-term studies. * The model was used ≥5 times.

## Data Availability

No new data were created or analyzed in this study.

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
