# Peer review of "The Application and Challenges of Brain Organoids in Exploring the Mechanism of Arbovirus Infection"

_microorganisms, 2025, doi:10.3390/microorganisms13061281_

Round 1
Reviewer 1 Report
Comments and Suggestions for Authors
- Although multiple studies are described, there is a lack of contrast between them: Which model is more representative? Which cell lines or culture conditions offer greater reproducibility? Suggestion: Add comparative tables of previous studies, including the virus used, organoid type, key findings, and reported limitations.
- Include a section discussing the model's epistemological limitations, specifically what questions organoids cannot answer.
- Although it is mentioned that organoids could be used for drug screening and vaccine evaluation, no concrete examples have been developed, nor have studies that have used this approach for arboviruses been cited.
Author Response
Comment 1: Although multiple studies are described,there is a lack of contrast between them: Which model is more representative? Which cell lines or culture conditions offer greater reproducibility? Suggestion: Add comparative tables of previous studies, including the virus used, organoid type, key findings, and reported limitations.
Response 1: We sincerely appreciate your thorough review of our manuscript and your insightful suggestions. In accordance with your suggestions, we have updated Table 1 to include virus used, organoid type, key findings, and reported limitations. To enhance clarity and conciseness, we used an asterisk (*) to mark brain organoid models that were employed more than five times in published studies. An additional supplementary table, Table S1.
Furthermore, in the revised manuscript (page 5, lines 172-180) , we have incorporated a new analytical paragraph to provide more comprehensive interpretation of the tabulated data, with these additions clearly highlighted in yellow for easy reference.The added content reads:"In studies of arbovirus infections (e.g., ZIKV, JEV, LACV), brain organoids derived from hESCs and hiPSCs are used equally often (hESC: 8 times; hiPSC: 9 times), with no preference for either cell type. Brain organoids featuring multiple brain regions are the most common model, using a culture system of DMEM/F12 with Neurobasal medium, N2/B27 supplements, and SMAD pathway inhibitors (SB-431542 or LDN-193189) with bFGF/EGF for neural induction. During maturation, neurotrophic factors like BDNF and GDNF, combined with a Matrigel 3D culture system, support synapse formation and neuronal maturation. Rotary [25,35,40,45] and swinging [37,39,46] oscillation cultures are preferred for better nutrient diffusion and oxygenation(Table S1)."
Table S1 Comparative analysis of brain organoid models for arbovirus research
|
Origin |
Types of brain organoid |
Virus |
Number of applications |
Key differentiation factors |
Culture method |
|
hESC |
Brain organoids (Cerebral organoids) |
ZIKV |
6 |
N2/B27 supplementation, SMAD inhibitor induces neural differentiation, bFGF/EGF promotes proliferation, BDNF/GDNF facilitates maturation, Matrigel provides 3D support |
1.Stir flask bioreactors 2.rotary culture 3.oscillatory culture |
|
hESC |
Cortical organoids |
ZIKV,JEV |
2 |
IWR-1/SB-431542 induces dorsal telencephalic differentiation, FGF/EGF promotes neurosphere proliferation, and BDNF/GDNF/IGF-1/NT3 drives neuronal maturation |
/ |
|
hiPSC |
Forebrain-specific organoids |
ZIKV |
2 |
ROCK inhibitors form cell aggregates, neural differentiation is initiated using SMAD inhibitors, WNT modulators (IWR/XAV), N2/B27 maintains growth, and neuronal maturation is promoted by BDNF/GDNF. |
1.rotary culture |
|
hiPSC |
Brain organoids (Cerel organoids) |
ZIKV,LACV |
7 |
N2/B27, SMAD inhibitor initiates neural differentiation, bFGF/EGF promotes neural precursor amplification, Matrigel provides 3D support, and neurotrophic factors such as BDNF/GDNF promote maturation |
1.oscillatory culture 2.rotary culture 3.Stir flask bioreactors |
Note:/: Not available
Comment 2: Include a section discussing the model's epistemological limitations,specifically what questions organoids cannot answer
Response 2: We are deeply grateful for your suggestion regarding the epistemological limitations of organoid models. We have added a subsection entitled ‘4.4 Epistemological Limitations’ to the section of Section IV, which is located on page 12 (lines 343-351), with the following content which is highlighted in yellow in the revised manuscript: for emphasis:"At present, brain organoids remain at a developmental stage analogous to that of a fetus and are not yet capable of accurately replicating the physiological and pathological characteristics of the adult brain. This limitation includes the inability to model the mechanisms of arbovirus infections, such as those caused by the Japanese encephalitis virus (JEV), within a mature nervous system. Furthermore, the absence of vascularization and an immune microenvironment in these brain-like structures impedes the investigation of cross-organ interactions, such as the brain-gut axis, and hinders the evaluation of systemic inflammation and immune responses on the nervous system."
Comment 3: Although it is mentioned that organoids could be used for drug screening and vaccine evaluation,no concrete examples have been developed, nor have studies that have used this approach for arboviruses been cited.
Response 3: We sincerely appreciate the reviewer's valuable observation regarding the limited application examples of brain organoids in arbovirus drug and vaccine development. Indeed, as the reviewer rightly noted, concrete case studies in this specific field remain scarce in current literature. To address this critical gap, we have incorporated an important regulatory update in the revised manuscript (Page 14, Lines 422-426, yellow-highlighted): "The U.S. Food and Drug Administration (FDA) made a landmark announcement in April 2025, formally endorsing organoid platforms and organoid microarray systems as novel drug screening tools. This regulatory milestone establishes their potential to replace animal models for systematic drug safety evaluation [77]."
Reviewer 2 Report
Comments and Suggestions for Authors
Baoqiu Cui and colleagues reviewed the application of brain organoids in exploring arboviruses infection. The matter of debated is of interest to understand the key factors of arboviruses infection and development of antiviral therapeutic target. We appreciated the work but according our point of view there are some points to be improved.
Main points
1- In the introduction should be more in deep described the potential association of some arbovirus to neurological infection/pathogenicity. This is foremost relevance to understand the topic of the use of brain organoids.
2- In the paragraph 2, the description of brain organoids development lack of some information regarding the method used. For examples, the type of differentiating factor that induce the brain organoids differentiation or/and the type differentiating support used, if any. The general reader need of these information to understand the matter of the review.
3- The table 1 reported several factors that should be described in the specific arbovirus application. For examples, the “key factors” and “elucidate the mechanism” should be in part treated in the Zika virus infection model (as that of the other arboviruses described in the text) in order to understand their relationship.
Minor points
1- Some citation should be up-dated (e.g: reference 1 should be changes with a more recent literature).
2- Figure 3 should be clearly stated. Some point are not clearly reported.
3- All typos should be corrected.
Comments on the Quality of English LanguageNone
Reviewer 3 Report
Comments and Suggestions for Authors
The manuscript by Cui et al. entitled “The application and challenges of brain organoids in exploring the mechanism of arbovirus infection” comprehensively reviews the brain organoid-based methodology for the study of flavivirus infections. The work focuses mainly on ZIKV infection, which is the virus for which most results are available, although they also present data with other viral systems with neurological symptomatology (JEV, for example).
The development of brain organoids obtained from human embryonic stem cells (hESC) or human induced pluripotent stem cells (hiPSC) is a tool that provides valuable information on the arbovirus infectious process. This is evident during the reading of this review. However, the ethical approach to the development of these models is missing in the section on challenges and limitations. I believe that the authors could address the limitations that are beginning to be raised in certain forums. Some articles published in this regard can be found in:
10.1016/j.brainres.2020.146653
https://doi.org/10.1186/s13287-022-02950-9
https://doi.org/10.3389/frai.2023.1307613
https://doi.org/10.1038/s44222-024-00236-8
https://doi.org/10.1017/S0963180123000233
This type of discussion would close the section on limitations beyond the merely technical ones, heterogeneity, immunodeficiency and short lifespan, which are addressed in depth by the authors.
As minor comment, the publisher should check the use of capitalization or spaces, which in some sentences are wrong.
Author Response
Comment1: The manuscript by Cui et al. entitled “The application and challenges of brain organoids in exploring the mechanism of arbovirus infection” comprehensively reviews the brain organoid-based methodology for the study of flavivirus infections. The work focuses mainly on ZIKV infection, which is the virus for which most results are available, although they also present data with other viral systems with neurological symptomatology (JEV, for example).
The development of brain organoids obtained from human embryonic stem cells (hESC) or human induced pluripotent stem cells (hiPSC) is a tool that provides valuable information on the arbovirus infectious process. This is evident during the reading of this review. However, the ethical approach to the development of these models is missing in the section on challenges and limitations. I believe that the authors could address the limitations that are beginning to be raised in certain forums. Some articles published in this regard can be found in:
10.1016/j.brainres.2020.146653
https://doi.org/10.1186/s13287-022-02950-9
https://doi.org/10.3389/frai.2023.1307613
https://doi.org/10.1038/s44222-024-00236-8
https://doi.org/10.1017/S0963180123000233
This type of discussion would close the section on limitations beyond the merely technical ones, heterogeneity, immunodeficiency and short lifespan, which are addressed in depth by the authors.
As minor comment, the publisher should check the use of capitalization or spaces, which in some sentences are wrong.
Response1: We extend our sincere gratitude for your insightful and constructive feedback on our manuscript. Your forward-looking advice regarding the ethical considerations of complementary brain organ research has significantly enhanced the integrity of our study. Following your guidance and the references you provided, we have added a new section titled "4.5 Ethical Controversy" (page 13, lines 352-372). The details of this section are highlighted in yellow in the manuscript.“Brain organoid models, despite offering significant advantages in elucidating the mechanisms of arbovirus neuroinvasion, continue to encounter numerous challenges and limitations. From a technical standpoint, current models possess inherent deficiencies, including heterogeneity of cell types, absence of vascularization and immune microenvironments, and limited culture duration. In terms of ethical considerations, this nascent field is confronted with several unresolved dilemmas. The ethical debate surrounding the use of hESCs and the complexities of obtaining informed consent for hiPSC samples, particularly in cases involving surrogate decision-making for cognitively impaired patients, establish significant ethical barriers to research[5,6],The transplantation of virus-infected cerebral organoids into animal models for the study of the blood-brain barrier presents significant ethical challenges regarding the 'humanization' of animals and heightens the biosafety risk associated with the potential cross-species transmission of pathogens[5]. With the progression of cultivation technology, the prolonged cultivation of brain organoids may raise contentious issues regarding the potential for consciousness and their moral status. Consequently, there is an urgent need to develop a rigorous scientific framework for evaluating the 'threshold of consciousness'[6-9]. Arbovirus research demands strict biosafety due to its high pathogenicity, while the genetic data in personalized organoids raises donor privacy concerns. These ethical issues, along with technical and translational medicine challenges, form the complex barriers in brain organoid research on arboviruses.”
Additionally, we have meticulously proofread the entire manuscript to ensure accuracy and standardization in language expression.
Round 2
Reviewer 1 Report
Comments and Suggestions for Authors
The article is ready for publication.
Reviewer 2 Report
Comments and Suggestions for Authors
The authors have substantially improved the text making it suitable for the publication.
Reviewer 3 Report
Comments and Suggestions for Authors
The authors have improved the manuscript and I consider it ready for publication.